# Understanding the Changes in Quality of Semi-Dried Rice Noodles during Storage at Room Temperature

**DOI:** 10.3390/foods11142130

**Published:** 2022-07-19

**Authors:** Wen Xiao, Yuqin Ding, Ying Cheng, Sili Xu, Lizhong Lin

**Affiliations:** 1National Engineering Research Center of Rice and Byproduct Deep Processing, Changsha 410004, China; xw1530402235@163.com (W.X.); cy16680120451@163.com (Y.C.); xusili0826@163.com (S.X.); llz30@csuft.edu.cn (L.L.); 2College of Food Science and Engineering, Central South University of Forestry and Technology, Changsha 410004, China

**Keywords:** semi-dried rice noodles, quality characteristics, microbial growth, cooking quality, moisture distribution

## Abstract

The changes in semi-dried rice noodles during storage at room temperature (25 ± 2 °C) in terms of microbial growth, cooking quality, color, textural properties, thermal properties, crystallinity, and moisture content, and moisture distribution was examined. Total plate count, cooked broken rate, cooking loss, and b* value increased, while rehydration ratio, L* value, and moisture content decreased during storage. The hardness, adhesiveness, and chewiness of semi-dried rice noodles increased significantly, according to textural properties. DSC and XRD showed that the enthalpy of thermal absorption and crystallinity of semi-dried rice noodles increased from 1.67 J/g and 3.48% to 4.21 J/g and 18.62%, respectively. LF-NMR showed that the weakly bound water content in semi-dried rice noodles decreased by 3.71%, and the bound water content and free water content increased by 3.20% and 0.51%, respectively. The results of correlation analysis showed that the changes in quality during storage of semi-dried rice noodles were influenced by the combination of microbial growth, aging of rice noodles, and moisture migration.

## 1. Introduction

Rice noodle is a widely consumed staple food in many Asian countries due to its unique flavor and texture [1,2]. Depending on the moisture content, rice noodles can be divided into fresh wet, semi-dried, and dried rice noodles [3]. Fresh rice noodles have excellent flavor and texture but have a limited shelf life due to their high moisture content. The flavor and texture of rice noodles may be diminished in the dehydration process, and it is often necessary to extend the shelf life of rice noodles. Nowadays, consumers are becoming more and more concerned with convenient and healthful food [4]. As a result, it is a good choice to reduce the moisture content of rice noodles to a relatively safe level, which is found in semi-dried rice noodles [5]. This guarantees a similar flavor and texture to fresh rice noodles and a comparable longer shelf life to dry rice noodles.

The storage quality of food has an impact on consumer acceptability as well as their health and safety, and it also has an impact on the producers’ reputation and profitability. Any food that reaches the consumer should be investigated for storage quality before entering the market [6]. Nowadays, adding additives, optimizing manufacturing conditions, or using advanced technological treatments are typically the best ways to improve the storage quality of rice noodles. Tantala et al. [7] indicated that the addition of 0.1% chitosan and 0.1% potassium sorbate to fresh rice noodles could effectively inhibit the growth of molds and microbes during storage. Bai-Ngew et al. [8] found that the addition of a moderate amount of crude peptide extract from lablab bean in semi-dried rice noodles was able to prevent the growth of Bacillus during storage and prolong the shelf life. Rachtanapun et al. [9] demonstrated that packing fresh rice noodles using a nylon pouch can improve the sensory quality. Low et al. [10] reported that soaking in gluconic δ-lactone (GDL) followed by pasteurization of fresh rice noodles can improve the quality significantly and delay the long-term aging. Shi et al. [11] reported that treatment of fresh rice noodles with electron beam irradiation (EBI) results in a significant reduction in the initial total bacteria counts and fungal counts (mold and yeast) and the extension of shelf life. However, there are few studies on changes in the storage quality of rice noodles. Yang et al. [12] demonstrated that the hardness and color decreased, and cooking loss increased significantly with increased storage time during storage of fresh rice noodles at 25 °C. Xue et al. [13] showed that the amount of mold and saccharomyces increased, acidification decreased, and cooking loss increased significantly with increased storage time during storage of fresh brown rice noodles. However, there are few studies on the quality changes of semi-dried rice noodles during storage, and supplementing this data gap is essential for improving the quality of semi-dried rice noodles and extending the shelf life of the technology.

Therefore, in this paper, we measured the changes in total plate count, cooking quality, color, textural qualities, thermal properties, crystallinity, moisture content, and moisture distribution at different storage times (0 d, 5 d, 10 d, 20 d, 40 d, 60 d, 90 d, 120 d, and 180 d) to understand the quality variations of semi-dried rice noodles. Meanwhile, the mechanisms involved in quality changes of semi-dried rice noodles during storage were investigated by correlation analysis.

## 2. Materials and Methods

### 2.1. Materials

Both indica rice and strain fermentation solution are provided by Jinjian Rice Industry Co., Ltd. (Changde, Hunan, China). The strain fermentation solution consists of a microbial fermentation system composed of lactic acid bacteria and yeast, with lactic acid bacteria concentration of 1.0 × 10^8^ to 1.0 × 10^9^ CFU/mL and yeast concentration of 1.0 × 10^4^ to 1.0 × 10^5^ CFU/mL. Sodium chloride (Sinopharm Chemical Reagent Co., Ltd., Shanghai, China), casein tryptone (Solebro Technology Co., Ltd., Beijing, China), yeast (Auboxing Biotechnology Co., Ltd., Beijing, China), and agar (McLean Biochemical Technology Co., Ltd., Shanghai, China) were of analytical grade.

### 2.2. Preparation of Fermented Rice Flour

Indica rice was put into the fermenter, and distilled water was added according to the ratio of 1:1.2 (*w*:*v*), followed by a 4% of strain fermentation solution, and it was then put into a constant temperature incubator (35 °C) for 18 h. After fermentation, the indica rice was washed, drained, then crushed, and rice flour was dried in a drying oven (50 °C) for 12 h until the moisture content was 10~12% and was then passed through 120-mesh sieves and stored at 4 °C.

### 2.3. Preparation of Semi-Dried Rice Noodles

Next 50.0 g of rice flour was weighed, 62 mL of distilled water was added, and the mixture was stirred well and then heated in a 95 °C water bath for 4 min to make the rice flour paste into a dough. The rice dough was extruded into strips by pasta machine (MS-200, Hualian Shengtong Trading Co., Taiyuan, Shanxi, China) to obtain rice noodles. Subsequently, the rice noodles were re-steamed for 3 min, cooled, and aged in the incubator (LRH-250, Yiheng Scientific Instruments Co., Shanghai, China) for 8 h. Finally, the semi-dried rice noodles were vacuum-packed and subsequently sterilized in hot water at 95 °C for 30 min and then quickly cooled in ice water to prevent microbial reproduction.

### 2.4. Storage Conditions and Sampling Time of Semi-Dried Rice Noodles

The prepared semi-dried rice noodles were stored at room temperature (25 ± 2 °C) for 180 days and sampled at specific times (0 d, 5 d, 10 d, 20 d, 40 d, 60 d, 90 d, 120 d, and 180 d) to determine the changes in the quality of semi-dried rice noodles during storage. In addition, the interval between sampling and unfolding of the experiment was no more than 2 h for semi-dried rice noodles.

### 2.5. Determination of Total Plate Count (TPC)

TPC of semi-dried rice noodles samples was determined according to GB/T 4789-2016 [14] (Code of National Standard of China, 2016). For the experiment, 25 g of samples were placed in 225 mL of 0.85% sterile saline and aseptically homogenized for 120 s. A series of tenfold dilutions (10^−1^ to 10^−5^) were prepared, and 0.1 mL of the dilutions in sterile saline were inoculated onto PCA (plate counting agar) medium. PCA medium was incubated at 37 °C for 48 ± 2 h.

### 2.6. Determination of Moisture Content and Distribution

The moisture content of semi-dried rice noodles during storage was determined using a moisture tester (JH-H5, Yixing Instrument Co., Ltd., Taizhou, Jiangsu, China) at a room-temperature environment (25 ± 2 °C).

The moisture distribution of semi-dried rice noodles was determined using a nuclear magnetic resonance imaging analyzer (NMI20, Newsmax Analytical Instruments, Suzhou, China). According to the method of Qiao et al. [15] with slight modifications, 10 selected semi-dried rice noodles samples of about 3 cm in length were placed in the NMR tube and sealed with cling film to prevent moisture evaporation. Parameters measured were SW of 200 kHz, TW of 2000 ms, TD of 150,828, NS of 32, and NECH of 2000.

### 2.7. Determination of Cooking Quality

#### 2.7.1. Cooked–Broken Rate Measurements

The cooked–broken rate of semi-dried rice noodles was determined by referring to the method of Zhu et al. [16] with slight modifications. We selected 20 semi-dried rice noodles of length about 20 cm, put them into a beaker with boiling water, steamed them for 2 min and 30 s, and then put them into a test sieve to drain for 5 min. Selected rice noodles with lengths more than or equal to 10 cm and less than 10 cm were weighed separately, and the cooked–broken rate was calculated according to the following formula.
(1)$$\mathrm{Cooked}–\mathrm{broken}\;\mathrm{rate}=\frac{m_{1}}{m_{1}+m_{2}}\times 100\%$$
where *m*_1_: weight of semi-dried rice noodles more than or equal to 10 cm in length after cooked; *m*_2_: weight of semi-dried rice noodles less than 10 cm in length after cooked.

#### 2.7.2. Cooking Loss Rate and Rehydration Ratio Measurements

The cooking loss and rehydration ratios of semi-dried rice noodles were determined by referring to the method of Li et al. [17] with slight modifications. The moisture content of semi-dried rice noodles samples with different storage times was determined and recorded as *m*. Ten selected semi-dried rice noodles samples of about 20 cm in length were weighed as *m*_0_, placed in a beaker with boiling water, steamed for 2 min and 30 s, drained for 5 min after cooking, and weighed for the wet weight of rice noodles as *m*_1_. Then, the rice noodles were placed in a drying dish at 105 °C to determine constant weight as *m*_2_. The cooking loss and rehydration ratios were calculated according to the following equations.
(2)$$\mathrm{Cooking}\;\mathrm{loss}\;\mathrm{rate}=1-\frac{m_{2}}{m_{0}\left(1\;-\;m\right)}\times 100\%$$
(3)$$\mathrm{Rehydration}\;\mathrm{ratio}=\;\frac{m_{1\;}-m_{0}}{m_{0}}\times 100\%$$
where *m*: moisture content of semi-dried rice noodles; *m*_0_: weight of semi-dried rice noodles before cooked; *m*_1_: weight of semi-dried rice noodles after cooked; *m*_2_: dry weight of semi-dried rice noodles after cooked.

### 2.8. Determination of Color

The color of semi-dried rice noodles was determined using a colorimeter (UltraScan PRO instrument, HunterLab Instruments Inc., Reston, VA, USA). Fifteen selected semi-dried rice noodles of length about 20 cm were arranged neatly and placed at the through-hole of the colorimeter for measurement, mainly referring to the L* and b* values.

### 2.9. Determination of Textural Properties

Textural properties of semi-dried rice noodles were determined using a texture analyzer (TA. XT, Stable Micro Systems, London, England). According to the method of Huang et al. [18] with some modification, we selected 20 semi-dried rice noodles of about 20 cm in length, steamed them in boiling water for 2 min and 30 s, and then placed them on a test sieve to drain for 5 min. Then, we selected 5 rice noodles of about 5 cm in length after steamed, placed them on a texture meter, and measured them by TPA mode. The measurement parameters were P/36 R probe, pre-test speed of 2 mm/s, mid-test speed of 1 mm/s, post-test speed of 2 mm/s, the compression ratio of 50%, trigger force of 0.04905 N, and compression interval of 3 s.

### 2.10. Determination of Thermal Properties

The thermal properties of semi-dried rice noodles were determined by differential scanning calorimetry (DSC 2000, TA Instruments, New Castle, DE, USA). According to the method of Xu et al. [19] with minor modifications, 3.0 mg of powder samples were weighed (Semi-dried rice noodles samples were dried at 50 °C for 5 h, ground, and passed through 80-mesh sieves) in an aluminum crucible, distilled water was added at a ratio of 1:2 (*w*:*w*), and the samples were sealed using presses. Semi-dried rice powder samples were then equilibrated at 4 °C for 12 h. For DSC determination, the empty crucible was used as control, and the temperature was increased from 30 °C to 120 °C at a heating rate of 10 °C/min for testing.

### 2.11. Determination of Crystallinity

The crystallinity of semi-dried rice noodles was determined using an X-ray diffractometer (D8 Advance, Bruker AXS Inc., Karlsruhe, Germany) coupled with Ni-filtered Cu-Kα radiation. According to the method of Ge et al. [20] with some modifications, the diffraction scan region was 3° to 50°; the voltage was set at 40 kV, the current at 40 mA, and the scan speed was 2°/min. Furthermore, semi-dried rice noodles samples with different storage times were dried at 50 °C for 5 h, crushed, and passed through 200-mesh sieves.

### 2.12. Statistical Analysis

Data were analyzed using SPSS 22.0 (SPSS Inc., Chicago, IL, USA) for methodological analysis (ANOVA), significance analysis (Duncan), and correlation analysis. Origin 2020 (Northampton, MA, USA) and GraphPad Prism 8 (GraphPad Software, San Diego, CA, USA) were used for graphing. Experimental data are expressed as mean ± standard deviation (*n* = 3).

## 3. Results and Discussion

### 3.1. Changes in Total Plate Count (TPC)

The changes in total plate count (TPC) of semi-dried rice noodles during storage are shown in Figure 1. TPC reflects the microbial growth of semi-dried rice noodles during storage [21]. As shown in Figure 1, with the storage time increasing from 0 days to 180 days, TPC of semi-dried rice noodles firstly increased then decreased and reached the highest values when the storage time was 120 days. The highest TPC was 265 CFU/g, and it was much lower than the level of TPC ≤ 10^5^ CFU/g in rice noodles during storage reported by Yang et al. [22], which indicated that it could effectively prevent the growth of microbes after thermal sterilization and vacuum packaging of semi-dried rice noodles. TPC significantly increased with the lengthened storage period, and this was likely caused by the fact that thermal sterilization did not completely eradicate the microbes present in semi-dried rice noodles; instead, the surviving microbes kept multiplying and growing throughout storage, leading to TPC significantly increasing. After storage for 120 days, TPC of semi-dried rice noodle decreased because microbial growth and multiplication would have reduced the oxygen content in the packing bags and prevented microbial development. Similar results were made reported by Ntzimani [23], who reported that microbial metabolism would have reduced the oxygen content of the vacuum-packed bags and inhibited microbial development.

### 3.2. Changes in Moisture Content and Distribution

Water is a key component in many physical and biochemical processes, particularly microbial development, which affects the quality and storage stability of semi-dried rice noodles [24]. The changes in moisture content of semi-dried rice noodles during storage are shown in Figure 2A. The moisture content of semi-dried rice noodles decreased dramatically with extended storage time and decreased from 55.76% to 42.57%. This might be because the semi-dried rice noodles would be consuming certain amount of water for microbial development and metabolism during storage. This supports the results of Xue and Qiao, who reported that fresh brown rice noodles’ moisture content decreased with longer storage [13,15].

The changes in moisture distribution of semi-dried rice noodles during storage are shown in Figure 2B. LF-NMR was used to analyze the moisture state, moisture migration, and moisture distribution in foods [13]. As shown in Figure 2, A_21_, A_22_, and A_23_ were the bound water, weakly bound water, and free water contents in semi-dried rice noodles, respectively. The weakly bound water content exceeded 90.50% over the storage period, indicating that the moisture state in the semi-dried rice noodles was primarily weakly bound water. A_21_ and A_23_ increased, whereas A_22_ dropped with lengthened storage times. A_21_ increased from 2.50% to 8.58%, A_23_ increased from 0.22% to 0.92%, and A_22_ declined from 97.27% to 90.50%. This might be the result of the aging and recrystallization of branched starch; it reduced the water-holding capacity of semi-dried rice noodles, and some bound water was converted to free water, leading to an increase in the free water content [25]. Additionally, the recrystallization of starch molecules caused the water molecules that had been wrapped by the amorphous region of the starch to progressively spread to the crystalline region, leading to an increase in the bound water content [26]. The decreased content of weakly bound water was probably due to the balance of water inside the semi-dried rice noodles, which caused the weakly bound water to be supplemented inside and outside and converted to bound and free water.

### 3.3. Changes in Cooking Quality

The changes in cooking quality of semi-dried rice noodles during storage are shown in Table 1. The three main metrics for assessing the cooking quality of semi-dried rice noodles are cooked–broken rate, cooked loss, and rehydration ratio [27]. During storage, the cooked–broken rate and cooking loss of semi-dried rice noodles both significantly increased, while the rehydration ratio slightly decreased. The cooked–broken rate increased from 2.06% to 7.60%, cooking loss increased from 4.47% to 10.43%, and the rehydration ratio decreased from 59.14% to 48.51% of semi-dried rice noodles during storage. This might be attributed to the disruption of the gel network structure of starch during storage due to the aging and re-crystallization of branched starch, leading to a low ratio of rehydration, easy breakage, and turbid soup in semi-dried rice noodles after cooked [28]. Furthermore, the aging of semi-dried rice noodles would have reduced the adhesion between starch molecules, causing the cooked–broken rate and turbid soup of semi-dried rice noodles [29].

### 3.4. Changes in Color

The changes in the color of semi-dried rice noodles during storage are shown in Figure 3. Consumer acceptance of the appearance of rice noodles is directly influenced by color [30]. Generally speaking, rice noodles with higher L* values and lower b* values are more sought-after by consumers due to their brighter, glossier appearance. Semi-dried rice noodles’ brightness value (L*) decreased, and their yellow-blue value (b*) increased as storage time increased. L* values decreased from 81.63 to 78.05, and the b* values increased from 8.42 to 14.10, respectively. This could be the result of the browning reaction, such as the Maillard reaction, that occurred during storage [31]. The increased b* values could be the result of non-enzymatic browning components formation at the time of the storage process, as obtained by Kumar et al. in a storage study of cereal-based foods [32].

### 3.5. Changes in Textural Properties

The changes in texture properties of semi-dried rice noodles during storage are shown in Table 2. Customers place a great important value on the texture of rice noodles, and hardness is a key factor in determining textural qualities [33]. The hardness, adhesiveness, and chewiness were increased in various degrees of semi-dried rice noodles during storage. With more time spent in storage, the hardness increased from 2283.93 g to 5179.45 g, adhesiveness from 23.12 g·s to 90.72 g·s, and chewiness from 1629.22 g to 3218.39 g. While during storage, there were typically no appreciable changes to the springiness, cohesiveness, or resilience. Studies have revealed that changes in the textural characteristics are closely related to the aging of rice noodles [15]. The increased hardness and viscosity of semi-dried rice noodles were caused by the recrystallization of branched-chain starch molecules and the water outflow from the inside of the noodles during storage [34].

### 3.6. Changes in Thermal Properties

The changes in thermal properties of semi-dried rice noodles during storage are shown in Table 3. The onset temperature (T_o_), peak temperature (T_p_), termination temperature (T_c_), and enthalpy of thermal absorption (∆H) were increased to different extents in semi-dried rice noodles during storage. The onset temperature (T_o_), peak temperature (T_p_), termination temperature (T_c_), and enthalpy of thermal absorption (∆H) were increased to different extents in semi-dried rice noodles during storage. With more time spent in storage, T_o_ went from 49.87 °C to 53.83 °C, T_p_ from 58.47 °C to 63.33 °C, T_c_ from 66.62 °C to 71.26 °C, and ∆H from 1.67 J/g to 4.21 J/g. According to studies, the degree of starch aging was correlated with the magnitude of the enthalpy of thermal absorption, and the magnitude of the enthalpy of thermal absorption could be used to determine the degree of starch aging [35,36]. Larger thermal absorption enthalpy meant that the aging of starch was to a greater extent. During the storage process, the enthalpy of thermal absorption increased significantly, which indicated that the aging degree of semi-dried rice noodles increased during storage.

### 3.7. Changes in Crystallinity

The changes in crystallinity of semi-dried rice noodles during storage are shown in Figure 4. All samples of semi-dried rice noodles exhibited typical B-type crystals with typical diffraction peak positions at about 17°, 20°, and 22°, which were typical characteristic peaks of starch aging. According to the results, the semi-dried rice noodles’ aging appeared as starch aging, and the crystal structure of the noodles had not changed. The distinctive diffraction peaks at 17° can be seen in the XRD patterns to be more intense and to have larger peak regions for semi-dried rice noodles, which suggested that the aging of semi-dried rice noodles was mostly the aging of branched-chain starch during storage [37]. Semi-dried rice noodles’ crystallinity considerably increased with increased storage time, increased from 3.48% to 18.62%, and indicated the increased aging degree of semi-dried rice noodles during storage, which was consistent with the changes of ∆H in the thermal properties.

### 3.8. Correlation Analysis

In the actual experimental process, we discovered that TPC, cooking quality, color, textural properties, thermal properties, crystallinity, moisture content, and moisture distribution could only be used to describe the quality of semi-dried rice noodles from a single aspect, not fully describing the characteristics of semi-dried rice noodles and failing to reflect the inner connection of the quality changes of semi-dried rice noodles during storage. As a result, the correlation analysis of each indicator was necessary for semi-dried rice noodles during storage. The correlation analysis of each indicator of semi-dried rice noodles during storage is shown in Figure 5.

TPC has been considered as a key factor in the quality changes of rice noodles during storage [38], while both ΔH and crystallinity reflect the aging degree of semi-dried rice noodles, and the degree of aging was positively correlated with ΔH and crystallinity [39]. As shown in Figure 5, TPC, ΔH, crystallinity, A_21_, and A_23_ were all highly significantly and positively correlated with cooked broken rate, cooking loss, b* value, and hardness, and they were highly and negatively correlated with rehydration ratio, L* value, and moisture content. These results indicated that microbial growth, aging degree of rice noodles, and moisture migration were important factors that affected the quality of semi-dried rice noodles during storage. Based on the above, we could conclude that the quality changes of semi-dried rice noodles during storage at room temperature were affected by the combination of microbial growth, aging of rice noodles, and moisture migration.

## 4. Conclusions

Storage of semi-dried rice noodles is a complex process and involves changes in microbial growth, aging of rice noodles, and moisture migration. In the process of storage, TPC gradually increased and was up to 265 CFU/g. Cooked–broken rate and cooking loss increased and the rehydration ratio decreased. The L* value decreased significantly, the b* value increased significantly, and the appearance acceptability decreased. The amount of moisture dropped from 55.76% to 42.57%. After being stored for 180 days, the semi-dried rice noodles’ hardness, adhesiveness, and chewiness increased to 5179.45 g, 90.72 g·s, and 3218.39 g, respectively. DSC and XRD showed that the crystal type was B-type crystals, and the crystal structure did not change during storage, and the enthalpy of thermal absorption and crystallinity increased from 1.67 J/g and 3.48% to 4.21 J/g and 18.62%, respectively. LF-NMR showed that the moisture state mainly existed as weakly bound water in semi-dried rice noodles. As the storage time increased, the content of weakly bound water decreased by 3.71%, and the content of bound water and free water increased by 3.20% and 0.51%, respectively. Finally, the correlation analysis demonstrated that the changes in the quality of semi-dried rice noodles were influenced by the combination of microbial growth, aging of rice noodles, and moisture migration during storage. We have laid the groundwork for future studies on prolonging the shelf life of semi-dried rice noodles in this study.

## Figures and Tables

**Figure 1 foods-11-02130-f001:**
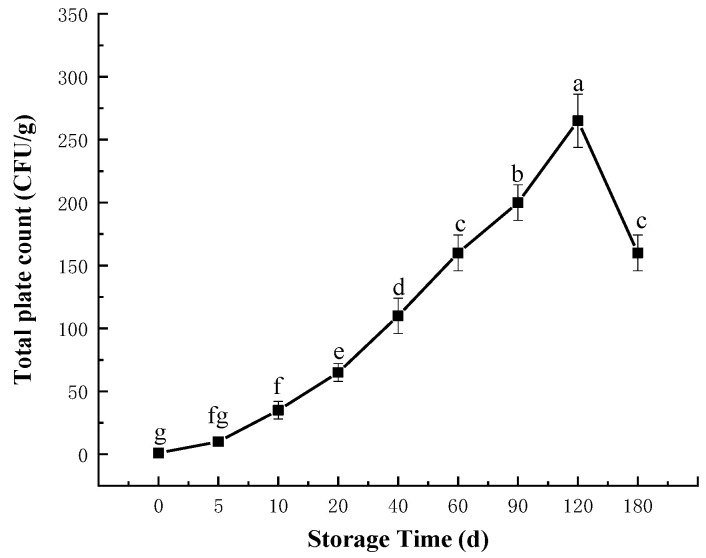
Changes in the TPC of semi-dried rice noodles during storage. Different letters denote significant difference (*p* < 0.05).

**Figure 2 foods-11-02130-f002:**
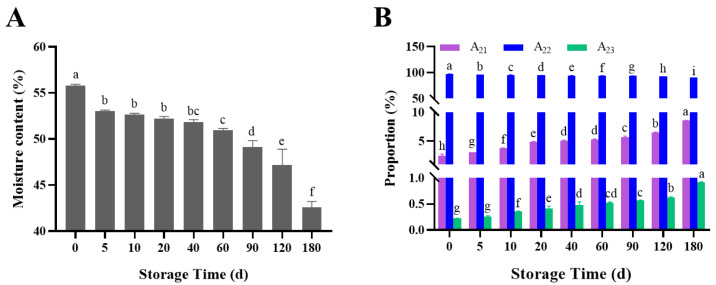
Changes in moisture content (**A**) and distribution (**B**) of semi-dried rice noodles during storage. Different letters within each color bar denote significant difference (*p* < 0.05).

**Figure 3 foods-11-02130-f003:**
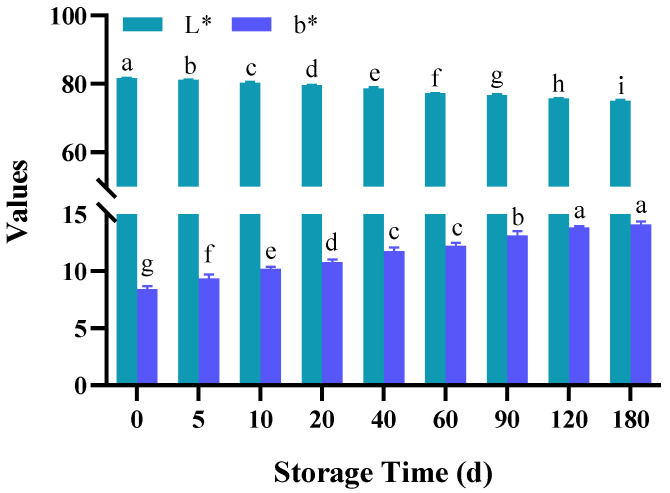
Changes in color of semi-dried rice noodles during storage. Different letters within each color bar denote significant difference (*p* < 0.05).

**Figure 4 foods-11-02130-f004:**
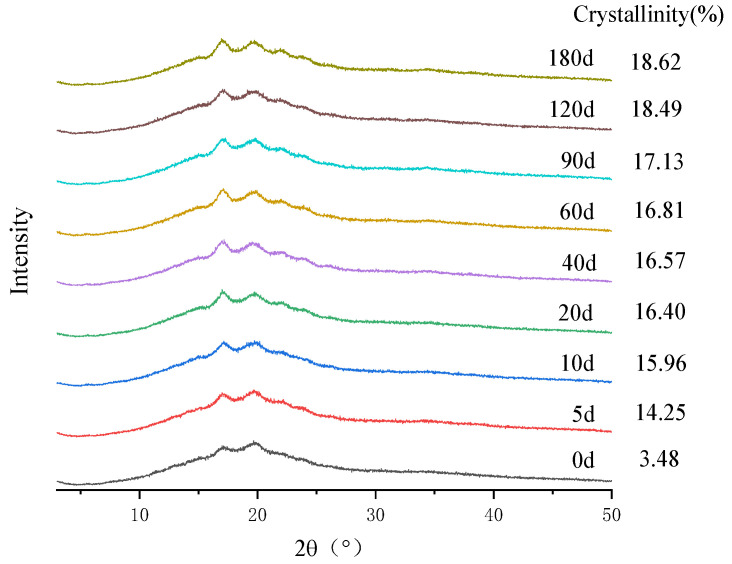
Changes in crystallinity of semi-dried rice noodles during storage.

**Figure 5 foods-11-02130-f005:**
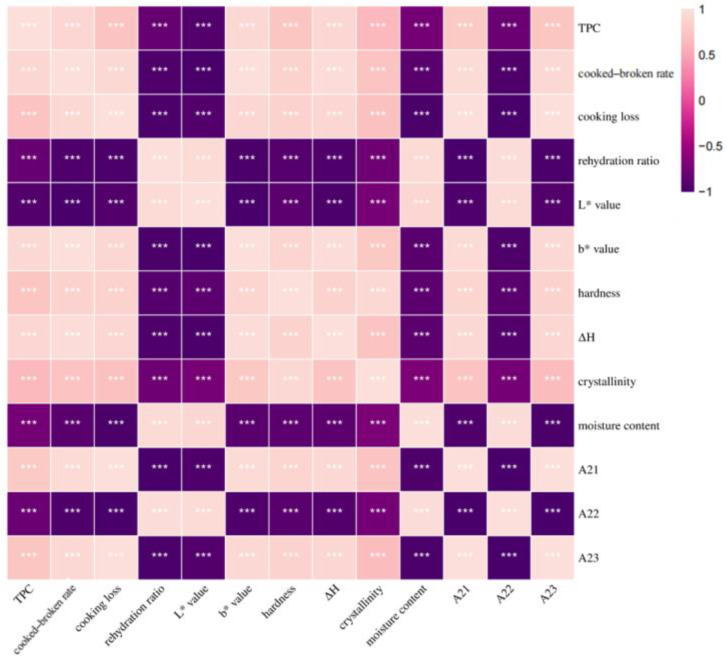
Correlation analysis of various indicators of semi-dried rice noodles during storage. Pearson’s correlation analyses were performed. *** *p* < 0.001.

**Table 1 foods-11-02130-t001:** Changes in cooking quality of semi-dried rice noodles during storage.

Storage Time (d)	Cooked–Broken Rate (%)	Cooking Loss (%)	Rehydration Ratio (%)
0	2.06 ± 0.10 ^h^	4.14 ± 0.25 ^g^	59.14 ± 0.52 ^a^
5	2.66 ± 0.27 ^g^	4.84 ± 0.18 ^f^	57.57 ± 1.11 ^b^
10	3.17 ± 0.08 ^f^	5.60 ± 0.27 ^e^	55.55 ± 0.20 ^c^
20	3.82 ± 0.36 ^e^	6.24 ± 0.24 ^de^	54.49 ± 0.22 ^cd^
40	5.33 ± 0.22 ^d^	6.62 ± 0.03 ^cd^	53.65 ± 0.15 ^d^
60	5.93 ± 0.14 ^c^	6.94 ± 0.14 ^c^	53.27 ± 0.12 ^de^
90	6.23 ± 0.11 ^c^	7.24 ± 0.09 ^bc^	52.35 ± 0.44 ^e^
120	7.15 ± 0.48 ^b^	7.83 ± 0.70 ^b^	50.62 ± 0.99 ^f^
180	7.71 ± 0.19 ^a^	10.43 ± 0.77 ^a^	47.84 ± 1.32 ^g^

The values are expressed as mean ± standard deviation of three replicates. There are no significant differences between means with identical letters in a column (*p* < 0.05).

**Table 2 foods-11-02130-t002:** Changes in textural properties of semi-dried rice noodles during storage.

Storage Time (d)	Hardness (g)	Adhesiveness (g·s)	Resilience (%)	Cohesion	Springiness (%)	Chewiness (g)
0	2283.93 ± 175.83 ^f^	23.12 ± 3.16 ^d^	41.57 ± 1.40 ^b^	0.76 ± 0.03 ^a^	93.37 ± 0.38 ^a^	1629.22 ± 106.28 ^e^
5	3643.42 ± 99.11 ^e^	36.35 ± 2.84 ^cd^	42.93 ± 3.05 ^b^	0.68 ± 0.02 ^ef^	89.68 ± 2.28 ^bc^	2222.82 ± 149.82 ^d^
10	3849.56 ± 29.06 ^de^	45.94 ± 0.63 ^bc^	48.41 ± 1.89 ^a^	0.70 ± 0.01 ^cd^	91.81 ± 1.52 ^abc^	2454.84 ± 31.45 ^cd^
20	4044.39 ± 37.73 ^cd^	64.81 ± 5.93 ^b^	49.06 ± 1.43 ^a^	0.70 ± 0.02 ^bcd^	89.23 ± 3.14 ^bc^	2530.73 ± 51.28 ^c^
40	4114.26 ± 86.64 ^c^	67.20 ± 7.28 ^b^	42.79 ± 2.25 ^b^	0.65 ± 0.03 ^f^	89.14 ± 0.34 ^c^	2376.20 ± 136.13 ^cd^
60	4295.04 ± 108.97 ^c^	60.24 ± 8.61 ^b^	52.40 ± 3.27 ^a^	0.73 ± 0.03 ^abc^	92.35 ± 0.70 ^ab^	2888.00 ± 109.72 ^b^
90	4226.20 ± 158.19 ^c^	101.12 ± 27.77 ^a^	53.48 ± 2.95 ^a^	0.75 ± 0.03 ^ab^	90.36 ± 1.93 ^abc^	2847.52 ± 216.74 ^b^
120	4742.12 ± 217.66 ^b^	53.94 ± 4.41 ^bc^	48.75 ± 2.48 ^a^	0.69 ± 0.03 ^cde^	90.40 ± 0.75 ^abc^	2961.39 ± 208.18 ^ab^
180	5179.45 ± 115.05 ^a^	90.72 ± 6.63 ^a^	48.43 ± 3.45 ^a^	0.68 ± 0.02 ^ef^	92.03 ± 1.22 ^abc^	3218.39 ± 109.60 ^a^

The values are expressed as mean ± standard deviation of three replicates. There are no significant differences between means with identical letters in a column (*p* < 0.05).

**Table 3 foods-11-02130-t003:** Changes in thermal properties of semi-dried rice noodles during storage.

Storage Time (d)	T_o_ (°C)	T_p_ (°C)	T_c_ (°C)	∆H (J/g)
0	49.87 ± 0.33 ^f^	58.47 ± 0.53 ^g^	66.62 ± 0.70 ^e^	1.67 ± 0.26 ^g^
5	50.93 ± 0.64 ^e^	60.54 ± 0.57 ^f^	67.33 ± 0.19 ^d^	1.77 ± 0.27 ^g^
10	50.72 ± 0.32 ^e^	61.34 ± 0.17 ^e^	69.05 ± 0.37 ^c^	2.41 ± 0.13 ^f^
20	51.90 ± 0.17 ^d^	61.62 ± 0.06 ^de^	69.59 ± 0.10 ^bc^	2.74 ± 0.11 ^e^
40	52.47 ± 0.06 ^cd^	62.05 ± 0.07 ^cd^	70.02 ± 0.37 ^b^	3.02 ± 0.03 ^d^
60	52.80 ± 0.08 ^bc^	62.44 ± 0.03 ^c^	71.08 ± 0.19 ^a^	3.41 ± 0.03 ^c^
90	53.09 ± 0.13 ^bc^	62.57 ± 0.21 ^bc^	71.30 ± 0.33 ^a^	3.71 ± 0.07 ^b^
120	53.13 ± 0.19 ^b^	63.00 ± 0.16 ^ab^	71.06 ± 0.37 ^a^	4.04 ± 0.04 ^a^
180	53.83 ± 0.59 ^a^	63.33 ± 0.20 ^a^	71.26 ± 0.45 ^a^	4.21 ± 0.03 ^a^

The values are expressed as mean ± standard deviation of three replicates. There are no significant differences between means with identical letters in a column (*p* < 0.05). T_o_, T_p_, and T_c_ were the temperatures of the onset, peak, and termination of thermal, respectively, and ΔH is the enthalpy change of thermal.

## Data Availability

Research data were not shared.

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
