# Peer review of "Understanding the Changes in Quality of Semi-Dried Rice Noodles during Storage at Room Temperature"

_foods, 2022, doi:10.3390/foods11142130_

Round 1

Reviewer 1 Report

This manuscript has evaluated the changes in the quality of semi-dried rice noodles during storage at room temperature. However, extensive editing of English language is required.

Many of the sentences are repeated several times or have similar meaning specially in the Introduction and Materials and methods. Moreover, the microbial population is very low but many of the obtained results is attributed to the microbial growth during storage. The discussion of must be improved and relevant references must be added.

 Comments:

Line 11: Change “We looked examined how semi-dried rice noodles changed” into “The changes in semi-dried rice noodles during storage at …”

Line 13: distribution was examined.

Line 73: … (Hunan, China). The strain …

Line 88: How much water was added to the rice flour?

Line 93: Sterilization temperature is higher than 100°C. Why you have used 95°C.

Lines 96-100: Paraphrase

Lines 103-104: sentence is not complete.

Fig 1: The TPC of semi-dried noodles is very low! Please recheck the data.

Fig 2: Why the free water is increased but the moisture content is decreased?!

Fig 3: The chart reveals that the rehydration ratio has not changed during storage but different days are shown with different letters. Please show the values in the table to show the differences.

Reviewer 2 Report

Very interesting and valuable work. A large number of different analyzes allows for the exact characterization of the raw material and recording changes resulting from the storage of noodles. In my opinion, the information on the number of repetitions in individual analyzes should be completed.

Line 268-269 - according to the results (Table 1), there was a deviation from the trend on the 40th day of storage - what could this be the result of?

Tables 1 and 2 please verify the correctness of the letter symbols (levels of significance of differences) in my opinion there are mistakes, especially Springiness and Tc

Line 334- "that" is repeated

Round 2

Reviewer 1 Report

the manuscript can be accepted